# Navigation and Control of Motion Modes with Soft Microrobots at Low Reynolds Numbers

**DOI:** 10.3390/mi14061209

**Published:** 2023-06-07

**Authors:** Gokhan Kararsiz, Yasin Cagatay Duygu, Zhengguang Wang, Louis William Rogowski, Sung Jea Park, Min Jun Kim

**Affiliations:** 1Department of Mechanical Engineering, Southern Methodist University, Dallas, TX 75275, USA; gkararsiz@smu.edu (G.K.); aduygu@smu.edu (Y.C.D.); zhengguangw@smu.edu (Z.W.); 2Applied Research Associates, Inc. (ARA), 4300 San Mateo Blvd. NE, Suite A-220, Albuquerque, NM 87110, USA; lrogowski@ara.com; 3School of Mechanical Engineering, Korea University of Technology and Education, Cheonan 31253, Chungnam, Republic of Korea

**Keywords:** magnetic manipulation, Microrobotics, non-Newtonian fluid, swarm control

## Abstract

This study investigates the motion characteristics of soft alginate microrobots in complex fluidic environments utilizing wireless magnetic fields for actuation. The aim is to explore the diverse motion modes that arise due to shear forces in viscoelastic fluids by employing snowman-shaped microrobots. Polyacrylamide (PAA), a water-soluble polymer, is used to create a dynamic environment with non-Newtonian fluid properties. Microrobots are fabricated via an extrusion-based microcentrifugal droplet method, successfully demonstrating the feasibility of both wiggling and tumbling motions. Specifically, the wiggling motion primarily results from the interplay between the viscoelastic fluid environment and the microrobots’ non-uniform magnetization. Furthermore, it is discovered that the viscoelasticity properties of the fluid influence the motion behavior of the microrobots, leading to non-uniform behavior in complex environments for microrobot swarms. Through velocity analysis, valuable insights into the relationship between applied magnetic fields and motion characteristics are obtained, facilitating a more realistic understanding of surface locomotion for targeted drug delivery purposes while accounting for swarm dynamics and non-uniform behavior.

## 1. Introduction

Microrobotics is a promising field for wide application areas such as medicine [1,2,3,4], space exploration [5], nanofabrication [6], microfluidics [7], biodefence [8], and environmental monitoring [9]. Considering the minuteness of the robots, which limits their manipulation abilities, achieving precise control and navigation of a microrobot becomes a formidable task. In order to manipulate the robots, researchers have utilized many techniques such as optical [10], electrostatic [11], acoustic [12], and magnetic actuation [13]. Magnetic actuation has advantages for microrobot manipulation, as it enables non-invasive control of the microrobot inside the body, offers high precision and control over movement, and is energy-efficient [14,15]. Additionally, magnetic actuation enables diverse motion capabilities for microrobots. By adjusting the magnetic field parameters, microrobots can exhibit translational motion, rolling, or even complex motion modes such as wiggling [16] or tumbling [17], depending on the design and magnetization configuration. With the advancement of manipulation techniques, it holds promise that these small robots will be capable of undertaking diverse tasks within restricted areas in the future, including potentially enabling minimally invasive surgery [18] and facilitating drug delivery [19].

The use of a magnetically actuated soft microrobot in a non-Newtonian fluid has important implications for the potential applications mentioned above [20]. Non-Newtonian fluids are common in biological systems (e.g. human body), and they have a different response to applied forces compared to Newtonian fluids [21]. Consequently, a non-Newtonian fluidic environment poses more complexities in forecasting the microrobot’s motion, as the behavior of non-Newtonian fluids tends to be more intricate and less predictable [22]. Given the specific environment in which microrobots are intended to be utilized, both the biocompatibility and biodegradability of a microrobot are important factors to consider for a range of tasks [23]. The microrobots can be fabricated by utilizing a biocompatible and biodegradable alginate material to form a soft deformable body [24]. When the contact between alginate solution and calcium chloride occurs, the soft microrobots can be created through crosslinking process [25]. To control the microrobots under external magnetic field, iron oxide particles can be incorporated into the alginate [26].

The soft alginate microrobots have been extensively studied in the literature for over a decade, showcasing significant advancements. For instance, [26] demonstrated the manipulation of spherical alginate particles using rotating magnetic fields. Another notable study by [27] utilized magnetic alginate-chitosan beads to achieve controlled release of insulin. Furthermore, ref. [28] presented a novel approach involving magnetically aligned nanorods encapsulated with alginate, which were successfully manipulated in a Newtonian fluid through tumbling motion. The objective in this paper was to comprehend the complex motion modes that arise in viscoelastic fluid environments. In our previous study [25], the locomotion of the microsnowman robot was examined only in a Newtonian fluidic environment using rotational magnetic fields to facilitate a rolling motion. In this work, by utilizing polyacrylamide (PAA), snowman-shaped soft microbots were deployed to demonstrate the feasibility of both wiggling and tumbling movements in non-Newtonian fluid. Both movements were predominantly influenced by the fluid’s viscoelastic properties, while wiggling motion could also be affected by non-uniform magnetization. Additionally, swarms of microbots with non-uniform behavior were exhibited within a complex environment. Those findings are helpful in optimizing the design and functionality of the microrobotic systems, ensuring their effectiveness in operating within such environments. The improvement in efficiency can also be pivotal in advancing the development of these systems for targeted drug delivery and non-invasive surgery. Ultimately, the use of soft microrobots in such applications has the potential to greatly improve the precision, efficiency, and safety of medical treatments, making it an exciting area of research with a significant potential impact on human healthcare.

The outline of the paper is structured as follows. Details of rolling and tumbling motions are explained in Section 2. In the Section 3, the wiggling and tumbling motions are showcased using a single robot and a swarm of microsnowman robots under a rotating magnetic field. A brief discussion regarding the results has been included in the Section 5.

## 2. Materials and Methods

### 2.1. Fabrication of Microsnowman and Preparation of Non-Newtonian Fluids

The microsnowman robots were fabricated from alginate hydrogels by an extrusion-based microcentrifugal method in Figure 1. The gelation process of alginate involves the transformation of a liquid alginate solution into a solid gel structure through a chemical reaction called cross-linking. Alginate powder is mixed with deionized water, to form an alginate solution. A separate solution, consisting of calcium chloride at a concentration of 5%, was prepared. The alginate solution is ejected in the form of a droplet by the centrifugal force from the tip of the needle and comes into contact with calcium chloride. The calcium ions bind to the carboxylic acid groups in the alginate chains, forming ionic bridges and creating a three-dimensional network [29]. This results in the formation of a gel structure with interconnected alginate chains [30]. The paramagnetic nanoparticles were embedded in the solution for encapsulation [25].

In experiments investigating the behavior of soft microbots in viscoelastic polymeric solutions, PAA, a water-soluble polymer, was employed. PAA is widely utilized in various industries, including wastewater treatment [31] and soil conditioning [32], due to its water solubility and diverse applications. The solutions were prepared by using PAA (Sigma Aldrich, Burlington, MA, USA, 92560) and added to deionized water in 0.25%, 0.5%, and 1% weight per volume (*w/v*) ratio [33]. As the concentration of the polyacrylamide (PAA) solution decreases below 0.25%, it starts resembling Newtonian fluids. This resemblance provides an interesting opportunity to examine the robot’s locomotion in these fluidic environments. However, once the concentration surpasses 1%, the fluid becomes highly viscous, making movement nearly impossible for the robot. The increased viscosity presents a significant challenge to its mobility. By investigating concentrations ranging from 0.25% to 1%, we aim to understand how different viscosity levels of the PAA solution affect the robot’s locomotion. A rheological measurement was performed after each solution was prepared. The measurement is accomplished using a rheometer [TA Instruments Discovery Hybrid (DHR-3)] attached to a 40 mm Peltier plate geometries disc (513400.905, H/A-AL ST SMART-SWAP). In the beginning, a series of calibrations are needed before the experiment, including geometry inertia and rotational mapping calibrations. After the calibration, a sample of one solution is added to the plate. When the measurement starts, a zero gap is applied on the contact surfaces between the equipment geometry and the sample solution, and the shear rate increases from 1 to 100 (1/s) with a 60-s step time. The total time to sweep over the share-rate range was 21 minutes. Measurements were made for 0.25%, 0.5%, and 1% PAA concentrations and were characterized by stress and viscosity concerning shear rates. The results in the logarithmic scale can be seen in Figure 2. It is clearly seen from the figure that there is a significant viscosity decrease with respect to the increase of shear rate for three concentrations of PAA. These results indicated that PAA has a shear-thinning pseudo-plastic property. Second-order curve fitting was applied to the raw data. The results of curve fitting can be seen in Table 1. The curve fitting presented in Figure 2 demonstrated a linear relationship between the sample’s viscosity and shear rate. In this case, the viscosity decreased linearly with increasing shear rate. The improving R-squared values suggested a better fit between the regression model and the data when the PAA concentration increases. For example, at 0.25% concentration PAA, the R-squared value was 0.8685, indicating 86.85% of the viscosity and shear rate data has a linear relationship. At 1% concentration PAA, the R-squared value increased to 0.9966, indicating that nearly all the data expressed linearity between viscosity and shear rate. Such results showed that the precision of such linearity between viscosity and shear rate improved when the PAA concentration increased.

### 2.2. Locomotion Methods

This paper explores the manipulation of the alginate microsnowman robot using tumbling and wiggling locomotion methods within a PAA solution. Rolling locomotion can be found in our previous paper [25]. As a result of the non-Newtonian fluid properties of the environment, such as uncertainty and non-linearity, the dominant factor affecting the propulsion of the microrobot’s location is the time-varying drag force. The thermal and fluidic effects such as Brownian motion were neglected due to the Reynolds number [34]. The highest calculated Reynolds number during our experiments is 0.07. Figure 3a,b illustrated the heterogeneous structure of microsnowman robots, consisting of two spheres of different sizes. However, it is important to note that due to the dimensional differences between the spheres, the applied torque and forces may undergo slight distortions. The overall applied torque (τ) on the microrobot can be calculated by,
(1)∑τ=M−F×r,
where *M* is the applied magnetic torque, *r* is the difference between the geometrical and dipole centers, and *F* is the drag force in 3D space. Because of the fabrication of microsnowman structures using the extrusion-based centrifugal method, the diameter sizes of each droplet for microsnowman configuration differ, which causes instability in the motion of the robot.

#### 2.2.1. Tumbling

Tumbling motion is mostly preferred for navigation in complex environments [35]. The higher radius of gyration and the ability to jump over the obstacles of tumbling motion enables the microrobots to effectively navigate and adjust to uneven terrains. To create the tumbling motion, the microsnowman robot rotates around its short axis [36]. The motion direction and rotation axis are aligned [37]. Figure 4 explains this phenomenon. The microrobot adapts its orientation and moves in various directions within complex environments by adopting a microsnowman configuration for tumbling. This enables it to overcome the conformational structures of the water-soluble PAA chain present in the fluid.

#### 2.2.2. Wiggling

There is a high shear force due to the non-Newtonian fluid behavior of PAA. Non-Newtonian fluids, like PAA solutions, exhibit viscosity that varies with the applied shear rate or stress (Figure 2). This means that the shear force experienced by objects moving through the PAA solution can be significant, depending on the flow conditions and the concentration of the PAA solution. As a result, we observed that the microsnowman robots moved in a constrained fashion even though the rotating magnetic field is applied with the same conditions. Additionally, the uncertainty associated with non-Newtonian fluids gives rise to this situation. Based on our experimental observations, we conclude that this motion resembles a wiggling motion. Figure 5 shows the steps of wiggling motion with a soft microsnowman microrobot on a substrate surface.

### 2.3. Experimental System

The experimental system is depicted in Figure 6a. The integrated electromagnetic coil system (MFG-100i, Magnebotix AG, Zurich, Switzerland) is installed on the inverted microscope (Nikon Eclipse TI, Tokyo, Japan). The power supply (ECB-820, Magnebotix AG, Zurich, Switzerland) was controlled by robot operation system (ROS) based software. A complementary metal-oxide-semiconductor (CMOS) color camera (Pixelink D734CU-T, Ontario, Canada, with resolution 2048 × 2048 pixels and 30 frames per second) via 0.5× Nikon C-mount and an objective lens (Nikon 2× Acromat) was used to capture video from the microscope on the bright-field mode.

The Magnebotix system consists of eight electromagnetic coils which are placed in a conical arrangement. The position of the electromagnets can be seen in Figure 6b. This configuration enables us to generate static and rotating magnetic fields as well as magnetic gradients in 3D space. By passing an electrical current through the coils and controlling the direction and magnitude of the current, a rotating magnetic field or magnetic gradient can be generated. The amplitude and direction of rotating magnetic fields with the electromagnetic coil system were generated by using the following formula,
(2)$$B=\left[\begin{array}{c} A\cos\phi \sin\theta \\ A\sin\phi \sin\theta \\ A\cos\theta \end{array}\right],$$
where *A*, ϕ, and θ represent the magnetic field magnitude (mT), azimuth, and inclination angles (degree), respectively. The angles and the amplitude is depicted in Figure 6c. The magnetic field was controlled by custom software with user input.

The velocity profile of the microsnowman robot was investigated to explore the motion characteristics in non-Newtonian fluidic environments. In this study, a frame-by-frame image processing algorithm was used to computationally track and calculate the velocity of microsnowman robots. The algorithm involved converting the captured images into binary images and subsequently locating the microsnowman robots, represented as white pixels on a black background. To determine the position of the microsnowman robots, pixel values were initially calculated and then converted to micrometers using a premeasured conversion value (1.54 μm/pixel) obtained from a grid micrometer glass slide observed under the microscope. By analyzing each frame, the position of the microsnowman robots as well as the distance traveled from the previous frame were determined. Additionally, by considering the camera’s frames per second (FPS) settings, which were set at 25 FPS, we were able to establish the time difference between frames. Leveraging all of these parameters, the instantaneous velocity of the microsnowman robots was accurately calculated throughout the experiments. These values were then averaged over each magnetic field set point.

## 3. Results

In this section, a series of experiments were performed using soft microsnowman robots to showcase their controllability within complex environments. Firstly, we conducted an analysis of the velocity profile in relation to the magnitude of the rotating magnetic field. Subsequently, the experimental results for microsnowman robots are presented within a swarm, focusing on their tumbling and wiggling motions.

The experiments were executed in a sample chamber in the center of the microscope, as seen in the center of Figure 6. The microsnowman robots were carefully selected and placed into the polydimethylsiloxane (PDMS) sample chamber (approximately, 5 mm diameter with 1.5 mm height). To prevent water evaporation and maintain fluid stability, a cover slip glass (25 mm × 18 mm, No 1.) was utilized as support. The PDMS chamber was prepared by mixing silicon elastomer (SYLGARD 184) and silicon elastomer curing agent (SYLGARD 1184) with a 7:1 weight ratio. Then, the mixture was heated on the hot plate to 60 ${}^{\circ }$C for 2 h.

### 3.1. Velocity Analysis with Microsnowman Robots in Non-Newtonian Fluids

Figure 7 shows the velocity versus applied magnetic field results of microsnowman robots in PAA solutions. The concentration of the solution in the experiments was 0.25%, 0.5%, and 1%, respectively. To ensure result consistency, the robot sizes were maintained within the range of 275–280 microns, while also maintaining a uniform density of iron oxide particles. For each concentration, attempts are made to keep robots at the same location for the experiments to eliminate the effect of fluid-structure interactions in the non-Newtonian fluid. For each magnetic field result, same rotating magnetic field is repeated for the same motion three times. To interpolate the results more clearly, a curve fitting is applied to the averaged velocities for the results of three experiments. The statistical analysis of the second-order fitting can be seen in Table 2. The average velocities increase with the use of a lower concentration of PAA. As a result of the shear-thinning characteristics of PAA, the velocity responses exhibited a gradual decrease in relation to the applied magnetic field for three different PAA concentrations. This effect is particularly evident in the case of 0.25% PAA. In higher shear rates, the loss of contact between the substrate surface and the microsnowman occurred as a result of molecular interactions in non-Newtonian fluids, leading to a slip phenomenon between the robot and the substrate. This situation caused higher errors after 5 mT was applied magnetic field. At the 10 mT applied magnetic field intensity, slippage was observed as a result of the deformation of the soft microsnowman body, leading to a decrease in velocity. In the case of 0.5% and 1% PAA concentrations, the predominant motion mode observed was wiggling. However, for the 0.25% PAA concentration, additional motion modes of tumbling and rolling were observed within the magnetic fields of 8–10 mT and 3–4 mT, respectively.

### 3.2. Swarm Control

Swarm control of microrobots involves the coordinated control of a substantial number of small robots, enabling them to perform tasks collectively [38]. Drug delivery using a swarm of microrobots requires greater control over the delivery process, which has the potential to enhance the effectiveness of the medication [39]. Through the synchronization of microrobot movements, it becomes feasible to administer drugs with heightened precision and accuracy. This coordinated approach ensures that the drugs reach their intended targets effectively while reducing the potential for side effects. Additionally, swarm control enables the delivery of multiple drugs simultaneously, which can improve treatment outcomes [40]. By using multiple microrobots to deliver different drugs to different targets, it is possible to create customized drug delivery regimens that are tailored to the patient’s individual needs. For these reasons, we attempted swarm control to understand the coordinated behavior of the microsnowmen in non-Newtonian fluids.

Figure 8 demonstrates the swarm control of microsnowman robots in non-Newtonian fluidic environments. Each robot was exposed to a uniform rotating magnetic field. At the lowest concentration, the robots showed the same wiggling motion modes. However, at the 0.5% PAA concentration, we observed a tumbling motion in the yellow path, whereas the other paths were manipulated with wiggling motion. PAA can exhibit heterogeneous characteristics. Due to its structure and composition, PAA can display non-uniform or varying properties across different regions or within a given system. Factors such as polymerization conditions, polymer chain branching, and the presence of impurities can contribute to the heterogeneous nature of PAA. Hence, the location has the ability to influence the environmental conditions, as demonstrated in Appendix A. During the microsnowman robot’s left turn along the yellow path, the dominant influence of additional drag force prevented the robot from achieving a tumbling motion. Additionally, the results revealed a decrease in the controllability of the microsnowman robots as the concentration of PAA increased.

Table 3 presents a comparison of the traveled distances between wiggling motion and tumbling motion in a 0.5% PAA solution, as observed in Appendix A. The table clearly shows that tumbling motion covers a greater distance when subjected to a uniformly rotating magnetic field signal. The disparity in performance can be attributed to variations in magnetization directions and geometries among the robots. In this context, single particles outperform microsnowman-shaped robots. This discrepancy arises due to the applied drag force, which negatively affects the motion of robots, especially those with larger surface areas like microsnowman robots, in contrast to single particles.

Figure 9 showcases the swarm control outcomes of the microsnowman robot, specifically focusing on the tumbling and wiggling motions. In Figure 9a, the effects of the PAA fluid’s uncertainty are evident. While the green path follows a square shape, the blue and white paths exhibit a rectangular pattern. This video presentation highlights the non-Newtonian behavior of PAA. Figure 9b demonstrates two distinct manipulation modes operating under the influence of the same global magnetic field.

The tumbling and wiggling motions are represented by the pink and red paths, respectively. This distinction arises from the varying dipole directions of the robots. If the direction of the alignment is along with the long axis, the tumbling motion is observed. On the other hand, the magnetic alignment is on the short axis, wiggling motion can be observed. This situation can be seen in Appendix A. Two heterogenous microsnowman robot manipulated with uniform global magnetic input exibit different motion characteristics. The robot following the pink path is magnetized along its long axis, while the robot in the red path is magnetized along its short axis. Figure 9 clearly illustrates that tumbling motion outperforms wiggling motion in terms of the traveled distance when subjected to a single magnetic input. The velocity calculation for tumbling motion yielded a value of 204.11 μm/s, while the velocity for wiggling motion was determined to be 127.86 μm/s.

## 4. Discussion

The presence of non-Newtonian fluids can significantly influence the drug delivery process involving microrobots. Non-Newtonian fluids exhibit complex flow behavior, where their viscosity and rheological properties are dependent on factors like shear rate, pressure, and composition. These fluid characteristics introduce challenges and considerations when it comes to the effective navigation and controlled release of drugs by microrobots. One effect of non-Newtonian fluids is increased resistance or drag encountered by the microrobots. The variable viscosity and flow behavior of non-Newtonian fluids can create higher levels of resistance, making it more difficult for microrobots to move through the fluid medium. This resistance can impede the overall mobility and maneuverability of the microrobots, affecting their ability to reach the targeted areas for drug delivery. Additionally, the complex behavior of non-Newtonian fluids can impact the release and dispersion of drugs carried by microrobots. The drug-loaded microrobots rely on controlled motion and precise positioning to deliver the drugs to specific sites within the body. However, non-Newtonian fluids may alter the flow dynamics and dispersion patterns of the drugs, making it harder to achieve targeted delivery. The varying viscosity and shear-thinning properties of these fluids can influence the drug release kinetics and alter the distribution of the released drugs. To overcome these challenges, researchers need to carefully consider the rheological properties of the specific non-Newtonian fluid encountered during drug delivery. Understanding the fluid’s behavior, such as shear thinning or shear thickening tendencies, can help in designing microrobots that can effectively navigate through the fluid medium and release drugs at the desired locations. By accounting for the complexities of non-Newtonian fluids, researchers can optimize the design and performance of microrobots to enhance drug delivery efficiency and precision in these dynamic environments.

## 5. Conclusions

In this study, the surface locomotion methods of microsnowman-shaped alginate microrobots were investigated within viscoelastic fluidic environments, with a focus on their applicability to real-world drug delivery scenarios. In comparison to swimming as a means of drug delivery, surface motions like rolling, tumbling, and wiggling offer advantages due to their improved controllability in various scenarios. Polyacrlamide (PAA) was utilized as a viscoelastic fluidic medium at concentrations of 0.25%, 0.5%, and 1%. Rheology analysis was performed specifically for the 0.5% and 1% PAA concentrations. The microsnowman robots were fabricated using an extrusion-based centrifugal method. Our objective was to investigate the complex motion modes that arise in viscoelastic fluid environments. By employing PAA, we deployed the microsnowman microrobots to demonstrate the achievability of both wiggling and tumbling motions. We found that wiggling resulted from the combination of viscoelastic fluid environments and non-uniform magnetization. Additionally, we showed that altering the viscoelastic properties can influence these motion behaviors, highlighting the non-uniform behavior of microbot swarms in complex environments. To characterize the motion in this complex environment, we conducted a velocity analysis against the magnitude of the applied rotational magnetic field. The robots were wirelessly controlled in an open-loop mode to track rectangular paths for each concentration of PAA solution. Tumbling and wiggling motions were achieved and compared both individually and in swarm configurations to see their feasibility for drug delivery in bodily fluidic environments. As a future direction, we plan to employ a nonlinear closed-loop controller in the experimental system. Another future work is to utilize a physiological fluidic environment such as blood or cerebrospinal fluid.

## Figures and Tables

**Figure 1 micromachines-14-01209-f001:**
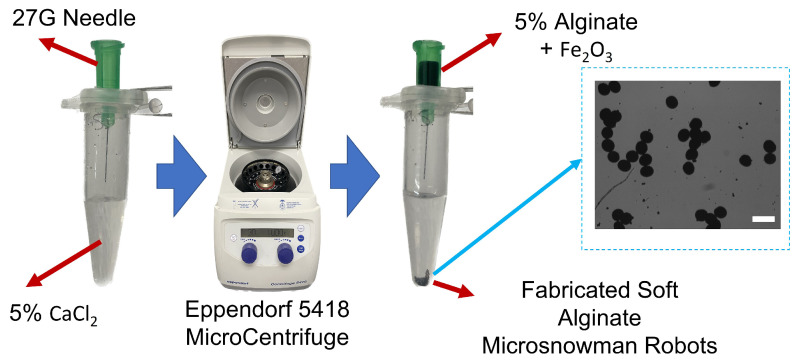
The fabrication scheme of soft alginate microsnowman robot. Calcium chloride (CaCl${}_{2}$, 5% *w/v*) was placed into a centrifuge tube (5 mL) with a sodium alginate (Na-Alginate, 5% *w/v*) in iron oxide solution (Fe${}_{2}$O${}_{3}$, 5% concentration, 50–100 nm particle size in diameter). The mixture was centrifuged with the microcentrifuge (Eppendorf 5418). Using the effect of centrifugal and gravitational forces, the droplets were generated and crosslinked into microsnowman robots. The generated particles are shown on the right side of the figure. The scale bar is set to 300 μm.

**Figure 2 micromachines-14-01209-f002:**
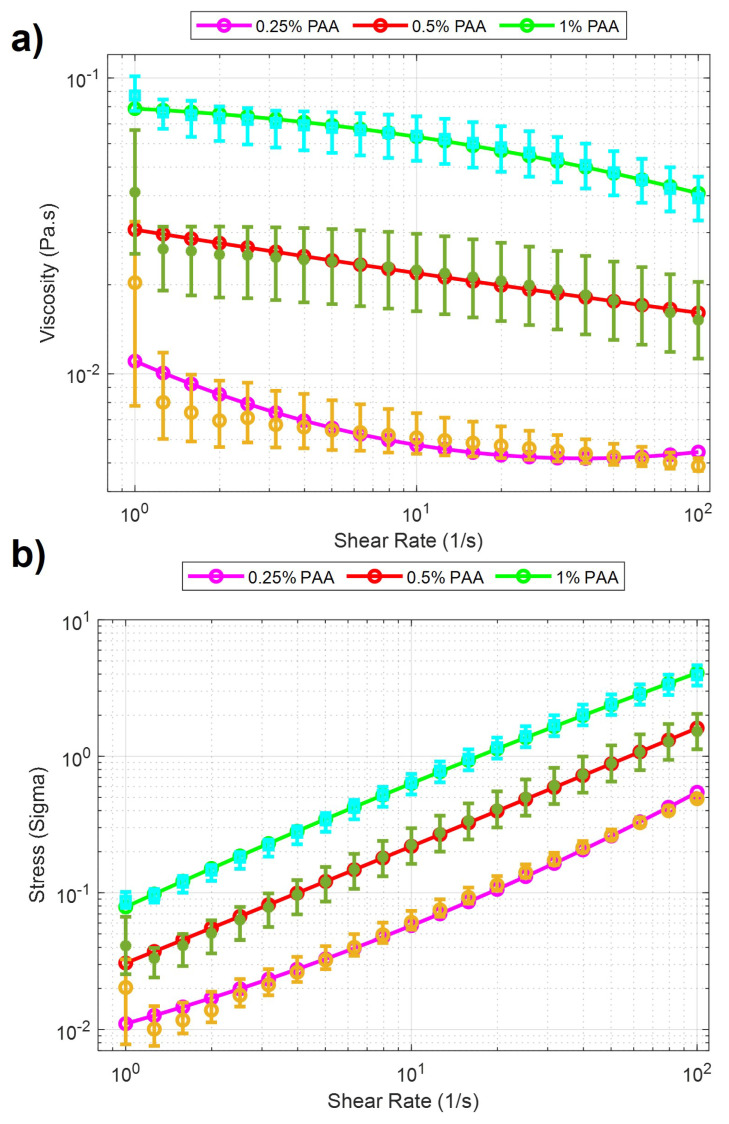
Rheology analysis results for PAA solutions. Purple, red, and green lines with cyan, green, and yellow error bars display the 0.25%, 0.5%, and 1% PAA concentrations. The data points from the characterization experiments are shown for three solutions, respectively. (**a**) Viscosity vs. Shear Rate. (**b**) Stress vs. Shear Rate.

**Figure 3 micromachines-14-01209-f003:**
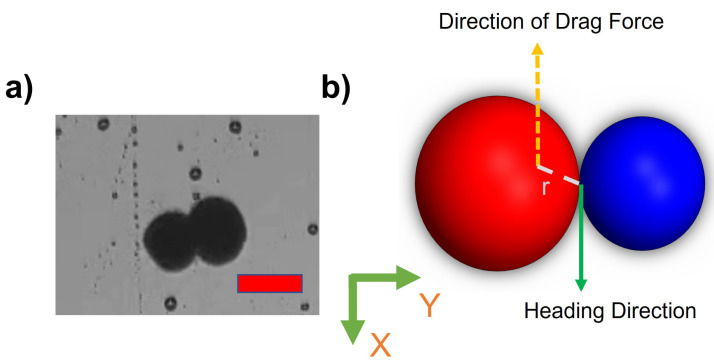
(**a**) An image of a heterogeneous microsnowman robot. The red scale bar is 250 μm. (**b**) The effects of imposed or applied forces in the 2-D plane. The north and south poles was shown in red and blue colors.

**Figure 4 micromachines-14-01209-f004:**
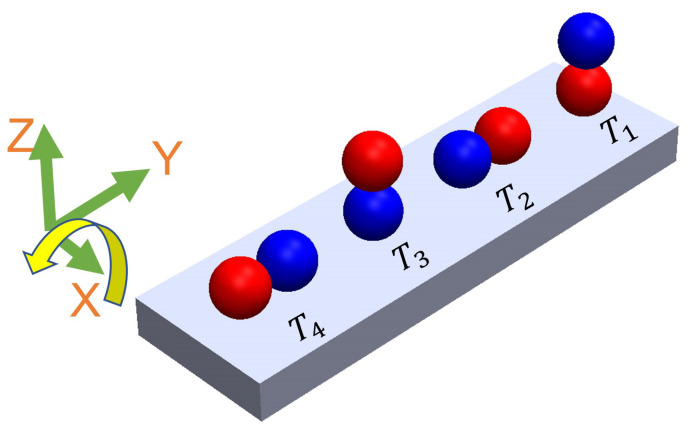
The steps of tumbling motion on a substrate surface. Red and blue colors represent the presumable north and south poles, respectively. The direction of the applied torque caused by the rotating magnetic field is shown by the yellow arrow. Each position of the robot was labeled from $T_{1}$ to $T_{4}$ with respect to time. The north and south poles are represented with blue and red colors, respectively.

**Figure 5 micromachines-14-01209-f005:**
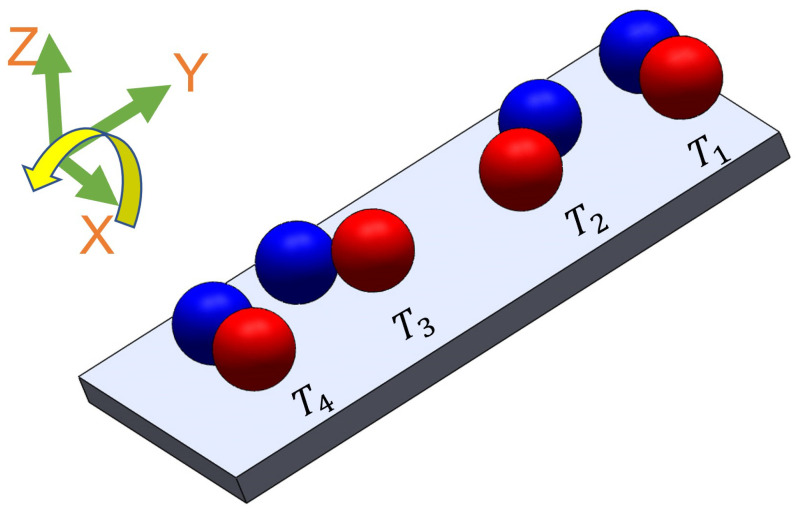
The illustration of wiggling motion on a substrate surface. Red and blue colors represent the presumable north and south poles, respectively. The direction of the applied torque caused by the rotating magnetic field is shown by the yellow arrow. Each position of the robot was labeled from T1 to T4 with respect to time. The north and south poles are represented with blue and red colors, respectively.

**Figure 6 micromachines-14-01209-f006:**
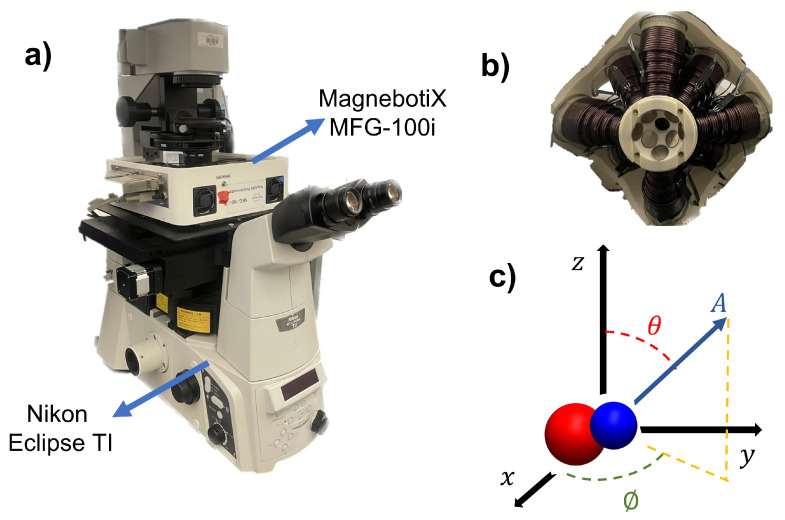
(**a**) Isometric view of experimental hardware. The magnetic field generator is mounted on the microscope. (**b**) The bottom view of the magnetic control system. The positions of the electromagnetic coils can be seen in the figure. (**c**) Schematic of the magnetic control system in 3D coordinates.

**Figure 7 micromachines-14-01209-f007:**
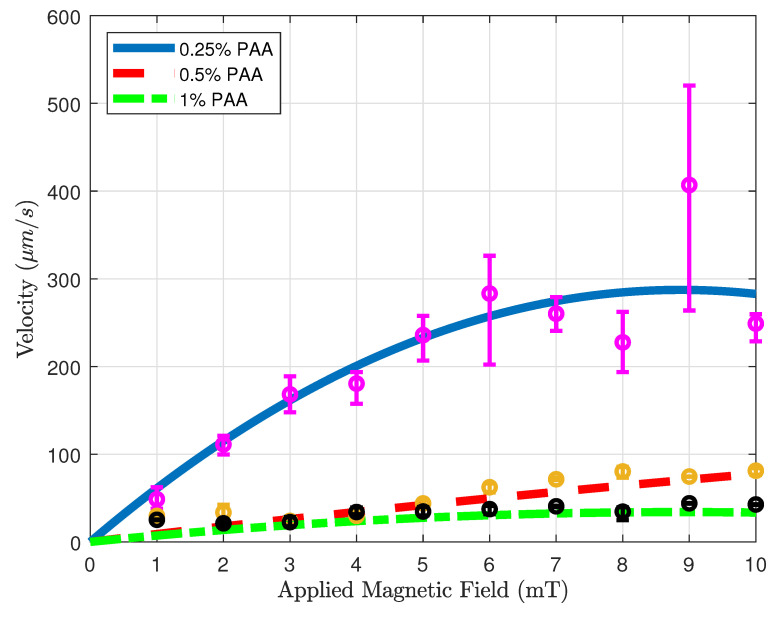
Velocity vs. magnetic field magnitude results for various concentrations. Blue solid, red dashed, and green dotted-dashed lines with the magenta, yellow, and black error bars represent 0.25, 0.5, and 1 % PAA solutions.

**Figure 8 micromachines-14-01209-f008:**
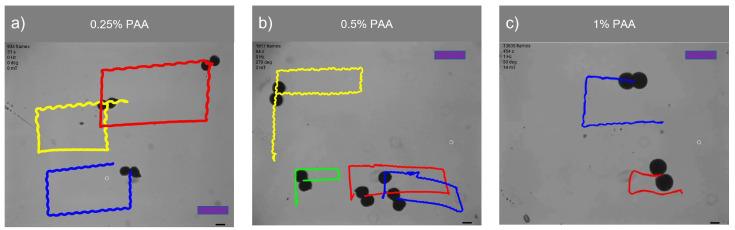
Swarm control of microsnowman robots under uniform global input. The scale bars represent 400 μm. The video can be found in the Appendix A. The frame numbers of each video are 925, 1954, and 13,779 frames per second for 0.25%, 0.5%, and 1% PAA concentrations.

**Figure 9 micromachines-14-01209-f009:**
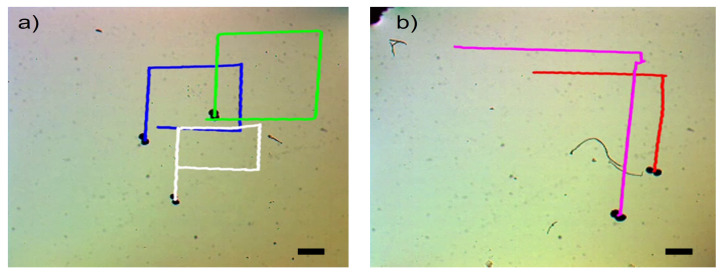
Complex locomotion in the non-Newtonian fluidic environment under single global rotating magnetic input. The scale bar is set to 500 microns. (**a**) Swarm control results for tumbling motion (See Appendix A). (**b**) Comparison of tumbling vs. wiggling motions. The pink and red paths are tracked for tumbling and wiggling motions (See Appendix A).

**Table 1 micromachines-14-01209-t001:** The curve fitting results for PAA characterizations.

**PAA Concentrations**	0.25%	0.5%	1%
**R-Squared Values (*R*^2^)**	0.8685	0.9295	0.9966

**Table 2 micromachines-14-01209-t002:** The curve fitting results for Velocity vs. Applied Magnetic Field.

PAA Concentrations	0.25%	0.5%	1%
R-Squared Values ($R^{2}$)	0.8495	0.8996	0.8476

**Table 3 micromachines-14-01209-t003:** Comparison for Distance and Velocity Global Uniform Magnetic Field.

	Tumbling (Yellow)	Rolling (Green)	Rolling (Red)	Rolling (Blue)	Single Particle
Distance (μm)	573.10	223.85	187.16	148.46	267.87
Velocity (μm/s)	37.21	14.53	12.15	9.64	17.39

## Data Availability

All data for this study have been experimentally generated and have been included in this paper.

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
