# Peer review of "Navigation and Control of Motion Modes with Soft Microrobots at Low Reynolds Numbers"

_micromachines, 2023, doi:10.3390/mi14061209_

Round 1

Reviewer 1 Report

This paper introduces the investigation of soft alginate microrobots motion characteristics with wireless magnetic fields in complex fluidic environments. The authors explore the various motion modes that arise as a result of shear forces in viscoelastic fluids. The velocity analysis investigated in this paper has provided valuable insights into the relationship between applied magnetic fields and motion characteristics, enabling a more realistic understanding of surface locomotion in the context of targeted drug delivery. The idea is interesting, and the developments are generally described clearly. Therefore, I suggest acceptance after addressing the following minor concern.

1.The authors mentioned “Microrobotics is a promising field for wide application areas such as medicine and environmental monitoring.”. The more state of art papers are suggested to add in the revised version, e.g., [1] C. Huang, Z. Lai, X. Wu and T. Xu. DOI: 10.34133/cbsystems.0004 [2] I. L. Sander, N. Dvorak, J. A. Stebbins, A. J. Carr and P.-A. Mouthuy. DOI: 10.34133/2022/9842169 [3] H. Zhou, G. Dong, G. Gao, R. Du, X. Tang, Y. Ma, et al. DOI: 10.34133/2022/9852853.

2.The diagram of how Tumbling and Wiggling perform would help the reviewers understand the motion type. The authors are recommended to display this in section 2.3.

3.How the magnetic field generate? The reviewer wants to know the detailed structure of the magnetic coil hardware, and how this system integrated with the microscope.

4.How these findings can provide guidance for microrobots application in targeted drug delivery. The authors are recommended to provide for discussion related to this.

Reviewer 2 Report

Gokhan Kararsiz and coworkers conducted a series of interesting studies about microrobot control in non-Newtonian fluid with low Reynold Number. Authors used PAA to simulate the non-Newtonian fluidic environment. They studied the wiggling and tumbling motions under various viscoelastic fluid environment. Moreover, they also discovered that swarms of microbots exhibit non-uniform behavior within complex environments. The manuscript is well written except a few comments below. I recommend the publication after addressing some minor concerns.

1. It is better if the signicance could be made more clear.

2. Please unify the use of nouns throughout the whole manuscript for better reading.(For example, wiggling and rolling)

3. Why did authors choose 0.25%, 0.5% and %1 concentrations of PAA for experiments? Could the motion Are the motion properties different at higher or lower concentrations? Please clear it in manuscript.

4. The conclusion is that tumbling is much faster than wiggling in distance and speed. Please describe clearly the control conditions for relevant motion.

5. Results demonstrate that the tumbling and wiggling occurred in 0.25% PAA.How does figure 5 count the speed?

6. Various motion characteristics will be obtained at different PAA concentration. I am concerning if the experiments could be conducted in different physiological fluid envoronment.(Such as blood or cerebrospinal fluid) It will be better if there will be these results.

Reviewer 3 Report

This paper investigates the effects of different fluid environments on soft microrobots of different composition concentrations (PAA concentrations). It showcases two different motion modes (wiggling and tumbling) and characterizes their speed and motion in different conditions. In general, the paper needs to provide more background in the field to place this work when compared to others before. Additionally, the drug delivery application are mentioned in the title and in many parts of the manuscript, but nothing is shown to support the claim that this microrobot is capable of it. I would suggest some sort of experiment that shows that the microrobot in question is able to hold and deliver a payload. Otherwise, make sure that it is clear to the reader that this is purely a paper on the motion characteristics of a magnetic microrobot.

Introduction:

- When introducing the microrobotics field, you only describe medicine and environmental monitoring as possible applications, making the field seem limited. More examples of application fields (with references) would be very beneficial.

- Similarly, when describing common microrobot actuation methods, it would be beneficial to add more references of other works to each actuation type. It might be also interesting to provide a very quick overview of each type and why magnetic is chosen for this work. You mention that it is because it enables wireless control, but other methods can provide actuation in a wireless manner as well. Some clarification here would be great.

- Starting on the second paragraph of the introduction, you are giving too much detail on what was done in this work, rather than give the reader more background on the field, what has been done, how this work is different, and why it is important. Most of the introduction can be placed elsewhere in the manuscript. The focus here should be introducing the work by giving the reader some background on it. Additionally, many important concepts are left without proper description. For example, thixotropy and rheopexy are mentioned but not explained. It should not be expected of the reader to go after the reference to find out what these terms mean.

- The end of the introduction section is good, but make sure to add at least some description of other similar works and show how this manuscript is different.

Materials and Methods:

- Please increase the font size of the labels in figure 1.

- Please include at least a summarized version of the fabrication procedure of the microsnowman.

- For figure 3, it would be more beneficial to have a larger version of the actual setup (inverted microscope + magnetic actuation system) with labels showing the important parts. Having the schematic of a computer and a CMOS camera doesn’t add much to the figure. Also, the coordinate axis can be included in the same schematic, saving some space for the image of the “real” setup.

- It would be more interesting to mention how the magnetic fields are applied (equation 1) after you describe the locomotion methods. Currently, it seems a bit out of place where it is in the text.

- Line 123 in the manuscript has a missing reference (?). Please address it.

Results:

- How was the velocity measured on a frame-by-frame basis? Would be interesting to at least mention the algorithm used to achieve these results. Also, how accurate are these measurements?

- In figure 5, is the curve fitting applied to the 0.5% robots (red line) the same as the other ones? It seems like it is linear and increasing in speed at the end rather than reaching a plateau (like the other two types of robots).

- In figure 6, there is no need to describe the length of the videos. It would be more interesting to add a description of the frames shown in the figure and their significance.

- It is not very clear in the text why the swarms move in different ways (and lengths) given the same global input. It seems to indicate that the impurities of the PAA are a contributing factor for it. Is this correct? If so, wouldn’t this be considered a severe drawback of this actuation method/microrobot swarm, since it would be more difficult to obtain a higher accuracy actuation?

- In the abstract, sentence flow can be improved. Too many very short sentences. Also, in the first sentence, make sure to note that the magnetic fields are used for actuation (make it clear for all types of readers). The use of very short sentences can be seen in other parts of the manuscript as well, which can be improved.

- Overall voice of the manuscript should be changed to third person, instead of first.

Round 2

Reviewer 1 Report

The authors have addressed all the comments raised by reviewers, I would recommend acceptance.

Reviewer 3 Report

Thank you for addressing my previous comments